# Higher Order Feature Extraction and Selection for Robust Human Gesture Recognition using CSI of COTS Wi-Fi Devices

**DOI:** 10.3390/s19132959

**Published:** 2019-07-04

**Authors:** Hasmath Farhana Thariq Ahmed, Hafisoh Ahmad, Swee King Phang, Chockalingam Aravind Vaithilingam, Houda Harkat, Kulasekharan Narasingamurthi

**Affiliations:** 1School of Engineering, Taylor’s University, 1, Jalan Taylor’s, Subang Jaya, Selangor 47500, Malaysia; 2Faculty of Sciences and Technologies, University of Sidi Mohamed Ben Abdellah, Route Imouzzer Fez, BP 2626, Fes 30000, Morocco; 3Faculty of Science and Technology, University of Algarve, Campus de Gambelas, 8005-139 Faro, Portugal; 4Simulation Metier, Valeo India Pvt Ltd., 1/396, Old Mahabalipuram Road, Navallur, Chennai, Tamil Nadu 600130, India

**Keywords:** gesture recognition, CSI, Wi-Fi, HOS, cumulants, mutual information, SVM

## Abstract

Device-free human gesture recognition (HGR) using commercial off the shelf (COTS) Wi-Fi devices has gained attention with recent advances in wireless technology. HGR recognizes the human activity performed, by capturing the reflections of Wi-Fi signals from moving humans and storing them as raw channel state information (CSI) traces. Existing work on HGR applies noise reduction and transformation to pre-process the raw CSI traces. However, these methods fail to capture the non-Gaussian information in the raw CSI data due to its limitation to deal with linear signal representation alone. The proposed higher order statistics-based recognition (HOS-Re) model extracts higher order statistical (HOS) features from raw CSI traces and selects a robust feature subset for the recognition task. HOS-Re addresses the limitations in the existing methods, by extracting third order cumulant features that maximizes the recognition accuracy. Subsequently, feature selection methods derived from information theory construct a robust and highly informative feature subset, fed as input to the multilevel support vector machine (SVM) classifier in order to measure the performance. The proposed methodology is validated using a public database SignFi, consisting of 276 gestures with 8280 gesture instances, out of which 5520 are from the laboratory and 2760 from the home environment using a 10 × 5 cross-validation. HOS-Re achieved an average recognition accuracy of 97.84%, 98.26% and 96.34% for the lab, home and lab + home environment respectively. The average recognition accuracy for 150 sign gestures with 7500 instances, collected from five different users was 96.23% in the laboratory environment.

## 1. Introduction

Device-free sensing implements human gesture recognition (HGR) using optical sensors [1], radio frequency (RF) [2,3] or ultra-wide bandwidth (UWB) [4] signals to achieve a higher gesture recognition accuracy. However, HGR using commercial off the shelf (COTS) Wi-Fi devices attracts significant research interest as it preserves privacy, is easy to deploy, low cost and readily available in an indoor environment. Wi-Fi based HGR systems capture signal reflections due to human movements between the transmitter and receiver pair as raw CSI traces. It has a wide range of application in the field of surveillance [5], physical analytics [6], healthcare [7] and have become a potential study in the smart home environment [8,9]. It is evident that the following factors, namely the number of users [10] and access point (AP) [11], orientation and distance between the users, as well as the transmitter and receiver pair [12,13], environmental factors [14], interferences, and multipath fading effect in the sensing environment influence the recognition accuracy [15].

Extensive researches were reported for feature extraction and selection, as it has a high impact on recognition accuracy of learning algorithms. Most of the existing recognition systems use Principal Component Analysis (PCA), a popular feature extraction technique to extract the principal components from the signal. For example, a CSI-based human activity recognition and monitoring system (CARM) [16] uses PCA for feature extraction and achieves a classification accuracy of 96% for eight different activities. The gesture recognition task initially acquires raw CSI traces of human reflections in subcarrier level with the implementation of the 802.11n standard in COTS Wi-Fi devices. The raw CSI traces consist of high-frequency noise and represents non-Gaussian signal distribution. Therefore, existing sensing methods pre-process the raw signal to reduce noise and apply transformations for unwrapping raw CSI measurement that reveals the phase change of the signal. The noise reduction phase mainly removes the phase offset, with outliers using the regression and filtering technique to de-noise the high-frequency signal. Fast fourier transform (FFT), inverse fast fourier transform (IFFT) and discrete wavelet transform (DWT) are the widely used signal transformation techniques for performing a linear transformation on the de-noised signal [17]. This pre-processed signal is of use in many applications that detect and locate human targets using CSI traces [18]. However, it pays little attention to addressing the linear phase error that prevails in CSI measurement, due to hardware limitations [19]. Ignoring useful information, such as the non-linear characteristic of the signal, will consequently affect the recognition accuracy in the later stage. Gesture recognition generally uses two approaches, namely deep learning (DL) and machine learning (ML). Recently, the research direction has migrated from traditional ML approaches to a DL approach, as DL methods report higher recognition accuracy. However, DL approaches demands a high volume of data for auto feature selection and better recognition accuracy and have a poor interpretation of data. Conversely, ML approaches can achieve satisfactory recognition accuracy even with a relatively lesser sample size but relies on the quality of hand-crafted features.

The limitations of the existing studies motivate the proposed work to study the non-Gaussian signal distribution with non-linearity, which carries useful signal information. The present work Higher Order Statistics based Recognition (HOS-Re) adopts a machine learning approach and proposes a HOS based third order cumulant feature extraction to maximize the recognition accuracy. The complexity of the classification model of the ML classifier increases with the number of feature inputs. Hence the present approach implements optimal feature selection methods that enable the machine learning algorithm to get trained faster and interpret the results easier by reducing the model complexity. Furthermore, it reduces overfitting and improves the accuracy of the model by mapping CSI traces to the ground truth values and by a selection of a robust subset of cumulant features. The idea proposed in the present work differs from the state-of-the-art method as it uses feature selection methods derived from information theory. Feature selection methods include mutual information feature selection (MIFS) [20], minimum-redundancy maximum-relevancy (mRMR) [21,22], conditional informative feature extraction (CIFE) [23], joint mutual information (JMI) [24], conditional mutual information maximization (CMIM) [25], double input symmetrical relevance (DISR) [26], interaction capping (ICAP) [27] and conditional redundancy (CONDRED) [28]. HOS-Re uses a secondary dataset, SignFi [29] for validating the proposed methodology. SignFi adopts a convolution neural network (CNN) [29] to measure the recognition accuracy. HOS-Re adopts multilevel support vector machine classifier that uses gaussian radial basis function (RBF) kernel.

In summary, the contribution of the paper is as follows:This paper proposes a HOS-based third order cumulant feature extraction from raw CSI traces and applies various feature selection methods derived from information theory, which effectively extracts and selects a robust feature subset considering the non-Gaussian signal distribution.HOS-Re measures the recognition performance using an ML classifier with feature inputs from feature selection methods. The experiment explores most of the mutual information based feature selection methods on cumulant features and identifies methods having higher recognition accuracy.To the best of our knowledge, this is the first reported work to use the combination of cumulant features + feature selection method for machine learning in the HGR domain using CSI traces and achieve significant performance.

In the remainder of the paper, Section 2 reviews the related work; Section 3 provides the preliminaries of this work. Section 4 gives an overview of the system under study, and details on feature extraction, selection, and classification. Section 5 discusses the implementation of the proposed methodology and evaluates the performance of HGR, and Section 6 concludes the discussion on HGR.

## 2. Related Work

Hand gesture recognition is an emerging research topic, especially for interpreting the sign language communication of people with speech and hearing challenges. The related works on gesture recognition are broadly classified under two major categories: (i) device-based gesture recognition and (ii) device free gesture recognition. Device-based gesture recognition widely uses wearable sensors like data gloves to capture hand gestures in order to interpret sign language [30,31]. However, device-free gesture recognition leverages signals of the commercial hardware devices for the gesture recognition task. These systems are categorized as: radar-based [32], RSSI [33,34] and CSI-based [15,35].

CSI is of interest as it helps to produce better action recognition accuracy since it consists of both amplitude and phase information. It provides fine-grained signal information in subcarrier level, which has widespread application in the field of computer vision. WiGer [36] adopts DWT for recognizing hand gestures in 5 different scenarios and achieves an average accuracy of 97.28%, 91.8%, 95.5%, 94.4% and 91% respectively. WiCatch [37] rebuilds the trajectories of the of nine moving hand gestures and achieves a recognition accuracy of 96% using SVM. WiFinger [33] recognizes eight different finger gestures with an average accuracy of 95% using PCA and DTW. WiKey [15] recognizes the keystroke of 37 keys of the keyboard using CSI with an accuracy of 77.4% to 93.4% with 30 and 80 samples per key respectively using PCA and DWT based feature extraction. Mudra [38] recognizes seven different finger gestures using a modified DTW algorithm with 96% accuracy.

State of the art CSI based sensing uses two major approaches: (i) using pre-processed CSI signals and (ii) using raw CSI signals. This section also discusses some of the works that adopt feature selection mechanisms.

### 2.1. Pre-Processed CSI Signals

Zhou et al. [39] uses PCA and achieves more than 97% detection accuracy with a localization error of 1.22 m and 1.39 m in a lab and meeting room environment using SVM. WFID [40] also applies PCA to uniquely identify humans from groups of nine and six people, and achieves an accuracy of 91.9% and 93.1% respectively with SVM. R-DEHM [41] achieves 94% accuracy in detecting human movements with an average error rate of 8% for duration estimation by implementing a back propagation neural network (BPNN) algorithm on principal components of CSI. R-TTWD [42] applies the majority vote algorithm on PCA based features to identify the presence and absence of human movement using one class SVM with true positive >99%. FallDeFi [43], a fall detection system achieves an accuracy of 93% and close to 80% in a pre-trained and new environment using PCA and SVM. WiFind [44] uses PCA based Hilbert-Hung Transform (HHT) to extract features for detecting the fatigue level of drivers and achieves accuracy of 89.6% and 73.9% for a single driver and multi-passenger, respectively, with one class SVM. Danger-pose detection system [45], detects a danger pose in the bathroom environment with an average accuracy of 96.23% using SVM as an anomaly detection algorithm. BodyScan [46] computes PSD from PCA features to recognize activities and detect breathing rates with 72.5% and 97.4% accuracy, respectively.

### 2.2. Raw CSI Signals

Few studies build a model-based recognition system that achieves better classification without signal pre-processing. For example, fine-grained real-time passive human detection, FRID [47], system computes the coefficient of temporal phase variation from the phase sensitivity due to human motion and detect the passive human motion with 90% precision. Gong et al. [48] proposed a threshold prediction model using Rician fading and the computation of cross-correlation factors to detect a human presence with a false positive and negative less than 4% and 5%, respectively. Gao et al. [49] proposed a deep learning model for implementing a device free wireless localization and activity recognition (DFLR) system using a sparse auto-encoder neural network. DFLR achieves recognition accuracy of 90% for eight different activities by transforming the CSI information as an image feature to the deep learning network. PriLA [50], the privacy preserving location authentication system, proposed a model exploiting the channel frequency offset and multipath information as the detrimental features and achieves an accuracy of 93.2% for authenticating user’s location information.

### 2.3. Feature Selection

This section reports some of the works that adopt feature selection paradigm. Wang et al. [51] used forward and backward feature selection to reduce the original 24 features obtained from the statistical data into 14 features. The feature selection with SVM reported better recognition accuracy as the selected features reveal the most useful information. WiHear [11] applies the multi-cluster/class feature selection (MCFS) algorithm to extract the optimal feature subset from the wavelet features. WiFi-ID [52] uses a combination of feature selection algorithm (relief) and sparse approximation-based classification (SAC) to extract the optimal feature subset and recognize the individual human subject.

## 3. Preliminaries

This section comprehensively discusses the basic concepts that are used in the proposed work.

### 3.1. Channel State Information (CSI)

CSI value reveals the channel properties of the communication link. Since the wireless medium is unstable and channel conditions may vary from time-to-time, channel estimation keeps changing on a short-term basis. COTS Wi-Fi devices following 802.11n standards work with orthogonal frequency division multiplexing (OFDM), achieve increased data rates, improved capacity, and reduced bit error rate (BER) of the system. Also, devices starting from IEEE 802.11n support multiple input multiple output (MIMO) with the OFDM scheme, enabling them to send and receive information over multiple antennas, as shown in Figure 1. The OFDM extracts the channel frequency response in the format of CSI, which contains both amplitude and phase information of the signal in subcarrier level, enabling the sensing to be more accurate. The CSI values at the transmitter and the receiver end may vary, and the data acquisition depends on how rapidly the channel conditions change. Therefore, the channel conditions highly influence the data acquisition of CSI traces and the instantaneous values need to be estimated in the receiver end on a short-term basis.

CSI contains information such as hardware timestamp, frame counter, number of receiving and transmitting antennas, received signal strength indicator (RSSI) of each antenna, noise, automatic gain control, amplitude, and phase information of the subcarriers in the form of a complex matrix. Among the information above, RSSI and CSI gain research interest in device free sensing methods. However, CSI is more preferred than RSSI, as RSSI does not provide the phase information. Equation (1) represents the received signal, which consists of signal information of the sender and the channel frequency response (CFR) with noise.
(1)$$R\left(f,t\right)=H\left(f,t\right)\;X\;T\left(f,t\right)+N,$$
where *R(f, t)* is the received signal strength of carrier frequency *f* measured at a time *t*; *H(f, t)* is the CSI in the form of CFR for carrier frequency *f* measured at a time *t*; *T(f, t)* is the transmitter signal strength for carrier frequency f measured at a time t and *N* is the noise. COTS Wi-Fi devices capture the varying signal characteristics of human reflections in the line of sight (LoS) or non-line of sight (NLoS) path between the COTS Wi-Fi device (Router) and AP’s (Laptop with Intel 5300 NIC) in the format of CSI, as shown in Figure 2. Wi-Fi signals from the transmitter get reflected from the floor, side walls, the ceiling, and objects in the confined experimental space.

### 3.2. Higher Order Cumulants

First and second order statistics have limitations to handle the non-Gaussian signal distribution, which may reveal some useful information contained in the signal. However, HOS addresses such a limitation by extracting higher-order cumulant information, and computes the unbiased estimate of variables. 

Let us consider a moment generating function mgXi exists for the variable *X*, whenever the statistical expectation, EeiX exists, *i*
∈ R. Therefore, the function gXi can derive the first, second, third and fourth order moments, *g*_1_, *g*_2_, *g*_3_, *g*_4_ as *mg*_1_*x* = *E* [*X*(*i*)]*mg*_2_*x*(*τ*_1_) = *E* [*X*(*i*) *X*(*i* + *τ*_1_)]*mg*_3_*x*(*τ*_1_, *τ*_2_) = *E* [*X*(*i*) *X*(*i* + *τ*_1_) *X*(*i* + *τ*_2_)]*mg*_4_*x*(*τ*_1_, *τ*_2_, *τ*_3_) = *E* [*X*(*i*) *X*(*i* + *τ*_1_) *X*(*i* + *τ*_2_) *X*(*i* + *τ*_3_)]
(2)
Let *τ*_1_, *τ*_2_, *τ*_3_ are the time lag parameters and let {*x*(*τ*)} be a *n*th order stationary random process, the *n*th order moment is the joint moments of orders up to *n* − 1 which is defined as:*mg_n_**x*(*ττ*_1_, *τ*_2_, *τ*_3_, …, *τ*_*n*−1_) = *E*[*X*(*i*) *X*(*i* + *τ*_1_) *X*(*i* + *τ*_2_) *X*(*i* + *τ*_3_) … *X*(*i* + *τ*_*n*−1_)]
(3)

The corresponding non-zero mean cumulants *C*_1_*x*, *C*_2_*x*, *C*_3_*x* and *C*_4_*x* can be computed as the non-linear combination of the moments as,*C*_1_*x* = *mg*_1_*x**C*_2_*x* (*τ*_1_) = *mg*_2_*x*(*τ*_1_)*C*_3_*x*(*τ*_1_, *τ*_2_) =*mg*_3_*x*(*τ*_1_, *τ*_2_)*C*_4_*x* (*τ*_1_, *τ*_2_, *τ*_3_) = *mg*_4_*x*(*τ*_1_, *τ*_2_, *τ*_3_) − *mg*_2_*x*(*τ*_1_) *mg*_2_*x*(*τ*_2_ − *τ*_3_) − *mg*_2_*x*(*τ*_2_) *mg*_2_*x*(*τ*_3_ − *τ*_1_) − *mg*_2_*x*(*τ*_3_) *mg*_2_*x*(*τ*_1_ − *τ*_2_),
(4)
and the *n*th order cumulant is derived using the *n*th order moment as:*C_n_x*(*τ*_1_, *τ*_2_, *τ*_3_, ⋯,*τ*_*n*−1_) = *mg_n_ x*(*τ*_1_, *τ*_2_, *τ*_3_, ⋯, *τ*_*n*−1_) – *mg_GN_ x*(*τ*_1_, *τ*_2_, *τ*_3_, ⋯, *τ*_*n*−1_),
(5)
where, *mg_n_ x*(*τ*_1_, *τ*_2_, *τ*_3_, ⋯,*τ*_*n*−1_) is the *n*th order moment and *mg_GN_x*(*τ*_1_, *τ*_2_, *τ*_3_, ⋯,*τ*_*n*−1_) is the *n*th order moment of the corresponding Gaussian process. The first order cumulant is the mean of the process while the second order cumulant is the autocovariance sequence, and the third is designed as the third central moment. The zero lag cumulants have special names: *C_2_x(0)* is the variance, *C_3_x(0, 0)* is the skewness and *C_4_x(0, 0, 0)* is the kurtosis. Third-order cumulants are preferred to extract the features from the raw CSI signal without applying any pre-processing. Fourth order cumulants build highly complex matrices, need huge computational efforts, and do not justify the recognition accuracy benefit over the efforts involved. Hence it is not considered in the present work. 

### 3.3. Feature Selection

Information theory feature selection algorithms measure the randomness of the feature *F* through entropy. The higher the entropy, the higher the uncertainty, which makes the feature become less informative. Feature selection methods derived from information theory measure the mutual information that exists between the features to reduce the uncertainty and select only the highly informative features [53]. The proposed work explores various feature selection algorithms using mutual information as relationship measure for computing a robust feature subset. The optimal feature subset will serve as the input to the machine learning classifier.

## 4. System Overview—HOS–Re

The proposed work emphasizes the use of non-linear behavior of raw CSI traces using a HOS cumulant approach. Figure 3 shows the schematic representation of the proposed HOS-Re methodology. The experimentation setup extracts the CSI values for the gestures performed by the users and stores the corresponding raw data with appropriate labelling to form a dataset. The presently proposed methodology extracts the higher order cumulant features from the raw CSI dataset using computational experiments. The feature selection step reduces the dimensionality of the extracted features and computes a robust set of best features. This feature subset will serve as a rationale input to the machine learning classifier and measure the performance of the system.

Figure 4 shows the detailed HOS-Re methodology adopted in the present work. Figure 4a shows that HOS extracts a set of third order cumulant features *F_n_* from raw CSI traces, as explained in Section 4.1. Information theory feature selection methods select a robust feature subset containing *k* mutually informative cumulant features from the extracted features *F_n_* as shown in Figure 4b. The values *F_a_*, *F_b_*, and *F_c_*, are chosen experimentally by feature selection algorithms to show the influence of feature subset on recognition, accuracy will be discussed in Section 4.2. The optimal subset of cumulant features serves as input to the ML classifier as in Figure 4c. The parameters are experimentally optimized in the classification step to estimate the recognition accuracy, as discussed in Section 4.3.

### 4.1. Feature Extraction

SignFi dataset is acquired in a scenario where background noises, signal interferences, and reflections exist in the presence of other Wi-Fi signals. Signal pre-processing techniques were deployed in most of the reported work, for improved recognition accuracy. When the signals are not pre-processed, accuracy declines. A statistical metric like power spectral density (PSD) [46,54] derived from the first and second order statistical moments, do not deal with the non-Gaussian signal distributions. The present work addresses this limitation by introducing third-order cumulant estimates for feature extraction and exploits useful signal information from the non-Gaussian distribution. Furthermore, third order cumulants consider amplitude, phase information, boost the signal to noise ratio (SNR) [55] and improve the recognition accuracy even without applying any pre-processing methods. Also, cumulant features are robust to background noises and signal interferences; hence, it is used in the present work. This section discusses third-order cumulant feature extraction from raw CSI traces in order to achieve better classification accuracy. 

#### Third Order Cumulants

HOS third order cumulant features are extracted from the raw CSI traces of the dataset with the assumptions that the statistical estimates are unbiased. Figure 5 shows the process of cumulant feature extraction. CSI data extraction from Wi-Fi signals needs a number of transmitting channels (*N_t_*) and receiving channel (*N*_r_). The receiving channel collects CSI measurements for the chosen sub carriers (*N_s_*), with a specific number of CSI samples per gesture.

The present work extracts 11 coefficients of the third order cumulant feature (*F*_1_ to *F*_11_,⋯, *F*_*n*−11_ to *F_n_*) from *N_t_* channels having *N_s_* subcarriers per channel. Thus, a total of *F_n_* cumulant features (= 11 × *N_t_* × *N_s_*) were extracted. Finally, the *n*th record of the cumulant feature removes the sample mean and stores only the unbiased sample estimates of cumulants, as discussed in Section 3.2. HOS-Re uses higher order spectral analysis (HOSA) toolbox integrated with MATLAB [56] for extracting the third order cumulant features.

### 4.2. Feature Selection

The HOS-Re extracts cumulant features from the raw CSI data. However, the recognition task with *F_n_* cumulant features is time-consuming and computationally expensive. In such a case, adopting a proper feature selection method simplifies the complexity of the recognition task by reducing the total number of cumulant features to form an optimal subset of features, as explained in Section 3.3. Information theory feature selection algorithms like MIFS, mRMR, CIFE, JMI, CMIM, DISR, ICAP, and CONDRED helps in reducing the *F_n_* cumulant features to *k* selected indices of feature subset based on mutual information. The feature selection criteria in the proposed work limit the feature size *k* as 30, 50 and 80 cumulant features to make it serve as well-grounded input to the multi-level SVM classifier to reduce the training time and complexity. The three values adopted in the present study are chosen from the experience gained in conducting some experimental runs to assess the performance of the present methodology. The results proved that if the set of the selected features is less than 30, the given accuracy drops significantly. When *k* is higher than 80, the recognition accuracy is stable with a marginal increase in recognition accuracy beyond *k* = 80, but the designed model becomes very complex. The ML classifier measures the performance of feature selection algorithms and identifies the best algorithm based on the recognition accuracy. HOS-Re implements feature selection algorithms, using feature selection toolbox (FEAST) integrated with MATLAB [28].

### 4.3. Human Gesture Recognition

The recognition task performed using an SVM with Gaussian RBF can non-linearly map samples into a higher dimensional space, unlike other linear kernels. The regularization parameter (C and γ) influences the classification performance of a non-linear SVM with Gaussian RBF kernel. C and γ are the parameters for a nonlinear SVM, which influences the accuracy and cost parameter of the resulting SVM model. SVM classifier is trained and tested with combinations of regularization parameter C and γ {0.003, 0.01, 0.03, 0.1, 0.3, 1, 3, 10, 30, 100 and 300} and validates the results using 10 × 5 cross-validation to find the optimal value of γ. The repeated cross-validation allows robust model assessment and reduces the variation in the model prediction to a greater extent. Experiments conducted using non repeated cross validation had shown that for some datasets, the results are completely unstable. When using a five cross-validation repeated five times, the results were found to be not very stable. Hence, a five cross-validation repeated 10 times is adopted in the present work. Multilevel SVM classifier with Gaussian RBF performs the multi-class classification problem using the feature subset of feature selection criteria. SVM partitions 80% of samples for training and 20% for testing. HOS-Re uses the library for support vector machine (LIBSVM) [57] package, which supports multiclass classification.

## 5. Implementation and Evaluation

This section discusses the implementation of the proposed methodology and performance evaluation of various feature selection method on the dataset identified from the literature. The present work uses raw CSI traces of the SignFi dataset [29] where the gesture movements refer to American Sign Language, ASL-LEX database [58], and the authors manually added labels for gestures that are not present in the database. 

### 5.1. SignFi Dataset

The authors of SignFi dataset [29] have captured the raw CSI measurements for 276 sign gesture with 20 and 10 instances for each gesture in the laboratory (lab) and home environments. The SignFi [29] dataset is chosen as the reference dataset for the present work, as this is the only dataset containing CSI traces with 276 gestures and a larger number of gesture samples with greater clarity in data acquisition. Hence, it is considered as the most promising dataset to implement our proposed methodology. There are 8280 gesture instances, 5520 from the lab and 2760 from the home, for 276 sign gestures in total from 1 user. Experiments also performed to capture CSI traces from 5 different users in the lab environment for 7500 instances of 150 sign gestures. 

The setting of the lab and home environment used for the measurement as reported in [29] is shown in Figure 6, and the description of the lab and home environments is reproduced here for immediate reference. The lab has more surrounding objects, leading to a more complex multi-path environment than the home. The distance between the Wi-Fi transmitter (AP) and receiver (STA) is 230 cm and 130 cm respectively, for the lab and home environment. For the home environment, the transmit antenna array is orthogonal to the direction from the transmitter to receiver. For the lab environment, the angle between the transmitter antenna array and the direct path is about 40 degrees. The major differences of these two environments are (1) dimension of the room, (2) distance between the transmitter and receiver, (3) angle between the transmit antenna array and the direct path, and (4) multi-path environments.

Table 1 shows a part of the full gesture labels representation of the SignFi dataset [29]. The raw CSI traces of the gestures were extracted using 802.11n CSI tool [59] in openrf [60] environment, with two Intel 5300 network interface card (NIC) equipped laptops. One laptop (AP) with three external antennas works as a transmitter (*N_t_* = 3), and another laptop with one internal antenna acts as a receiver (*N_r_* = 1) for data acquisition. Both transmitter and receiver operate at 5 GHz, with the channel width of 20 MHz in the Non-Line of Sight (NLoS) scenario. However, the 802.11n CSI tool [59] extracts the signal information for first 30 subcarriers (*N_S_* = 30) only, out of 52 subcarriers available, with 200 CSI samples per gesture. Interested readers are recommended to refer the SignFi [29] paper for more insight about data acquisition and experimental setup.

### 5.2. Cumulant Feature Extraction

HOS third order cumulant features extracted from the raw CSI traces of the SignFi dataset assumes statistical estimates are unbiased. The process of cumulant feature extraction shown in Figure 7, extracts the cumulants as feature vector. As explained earlier, the SignFi dataset based on Wi-Fi CSI signals deploys one transmitter with three internal antennae (*N_t_* = 3), one receiver (*N_r_* = 1) and collected 200 samples per gesture from 30 subcarriers (*N_s_* = 30).

HOS-Re, extracts 11 cumulant features (*F*_1_
*to F*_11_, …, *F*_331_
*to F*_341_, …, *F*_661_
*to F*_671_, …, *F*_980_
*to F*_990_) from 3 channels having 30 subcarriers per channel; thus a feature vector of 990 cumulant features (11 × 3 × 30 = 990) were extracted. Similar features were extracted from the dataset, which consists of 7,500 instances from five different users for 150 gestures with 200 CSI samples per gesture. Finally, the nth record of the cumulant feature removes the sample mean and stores only the unbiased sample estimates of cumulants, as discussed in Section 3.2. The total number of cumulant features extracted (990) should be optimally reduced based on certain feature selection criteria, to reduce the computational complexity.

The contours in Figure 8a–d shows 2D plots of the estimated third order unbiased cumulants, between the first-time lag and the second time lag. The plot for chosen gestures ‘HORSE’, ‘MONKEY’, ‘SNAKE’, ‘TIGER’ with overlapping segments, reveals the basic symmetry of the gestures. The time lag variable discussed in Equation (4), Section 3.2, with the variable maxlag set to 10 for revealing the basic symmetry in the plots.

### 5.3. Feature Selection

As mentioned in Section 4.2, several feature selection methods evaluated in the present study selects the feature indices *k* (30, 50 or 80) experimentally, to show the influence of sample size on accuracy. The feature selection algorithms that use mutual information as a relationship measure are explained in Table 2. Among the different feature selection algorithms evaluated, MIFS and CIFE are explained in the following section as they achieve better recognition accuracy and the details are discussed in the subsequent sections.

#### 5.3.1. MIFS

MIFS [20], a greedy feature selection approach, measures the mutual information between the candidate features and the output class, which reduces the uncertainty of preserving the useful information. This reduced feature subset serves as an optimal feature subset for the multi-class classification problem. MIFS is advantageous over linear transform dependent methods as it considers the non-linear relationship within the features and that between the features and the output gesture class. Therefore, MIFS provides a good generalization for the training and reduces the computational time. MIFS ensures that every selected feature is an informative one as follows: Set CF ← Initial ‘n’ cumulant features (*Fn* CF), S ← Empty setFor each cf ∈ CF, compute MI (GC; cf)(Compute mutual information, MI for every individual feature ‘cf’ that belong to the output gesture class ‘GC’)Find the cf that maximizes MI (GC; cf)Set CF ← CF\{cf}; Set S ← {cf}Greedy selection repeats until |S| = *k* (where, *k* = 30/50/80)Computation of MI between CF:for all pair of variables (cf, s) with cf ∈ CF, s ∈ S. Compute I(cf, s), if it is not already availableSelection of next CF:Choose feature cf as the one that maximizes I(GC; cf) – β $\sum_{s\in S}I\;\left(\mathrm{cf};s\right)$Set CF ← CF\{cf}; set S∈ S ∪ {cf}Output the set S containing the selected features (Selected feature subset with 30/50/80) 
where CF is the cumulant features, S is the set of currently selected indices of CF (30, 50 or 80) and β is the regularization parameter. MIFS ensures the term I(GC; cf)consist of only relevant CF and adds the CF iteratively in the feature subset S. Experiments were conducted by varying the β values from 0.1 to 1 as it contributes in reducing the redundancy among the CF.

#### 5.3.2. CIFE

The entropy of cumulant feature CF is denoted by H(CF) to measure the uncertainty and represent total information carried by the CF. Mutual information I (CF; GC) is defined by:
I (CF; GC) = H(CF) − H (CF|GC),
(6)

CIFE organizes an optimal feature subset that maximizes the class-relevant information between the features by reducing the redundancies among the cumulant features [23]. Equation (6) indicates the information delivered from CF to GC will reduce the uncertainty of GC when ‘CF’ is known. CIFE computes the joint class relevant information for k selected indices (30/50/80) of cumulant features as shown in Equation (2).
(7)$$I\;(\mathrm{CF};\;\mathrm{GC})\;=\;\sum_{t=1}^{n}I(\mathrm{CF}(t);\;\mathrm{GC})\;-\;\sum_{u=1}^{t-1}R_{\mathrm{GC}}(\mathrm{CF}(u);\;\mathrm{CF}(t)),$$
Equation (7) extracts the cumulant features sequentially given that (t − 1) CF are extracted, the tth feature can be extracted by optimizing the conditional informative objective as,
(8)θt= argmaxθtI CFt ; GC− ∑u=1t−1RGC(CF(u); CF(t)),
where θ is the tth CF parameter and Equation (8) derives the conditional informative cumulant feature.

### 5.4. HGR Performance Evaluation

HOS-Re evaluates the recognition performance of the proposed methodology using the machine learning classifier SVM with Gaussian RBF Kernel. SVM classifier evaluates the performance of feature selection algorithms with the combinations of regularization parameter *C* and γ. *C* is the cost parameter, and γ is the non-linear classification parameter of the Gaussian kernel. One value amongst {0.003, 0.01, 0.03, 0.1, 0.3, 1, 3, 10, 30, 100, 300} is assigned to *C* at a time and γ takes all values in sequence for each iteration, making 11 × 11 = 121 combinations and validates the results using 10 × 5 cross-validation. The recognition accuracy reported for the lab, home, lab + home dataset with 276 gesture and lab data set with 150 gestures using MIFS (*k* = 30/50/80 cumulant feature) by varying *β*.

The comparison of recognition accuracy of all feature selection algorithms like MIFS, mRMR, CIFE, JMI, CMIM, DISR, ICAP and CONDRED (*k* = 30/50/80 cumulant feature) for the lab, home, lab + home environment datasets are discussed below.

#### 5.4.1. Recognition Accuracy

Multilevel SVM RBF classifier measures the performance of the proposed model using accuracy as a metric, with selected *k* features fed as input from the feature selection algorithm. Figure 9a–d compares the performance of various feature selection algorithms with *k*, for the lab, home, lab + home datasets with 276 gestures and lab dataset with 150 gestures respectively. The overall accuracy increases with an increase in *k* for MIFS and CIFE, which reports higher recognition accuracy than other feature selection algorithms. 

Based on the results, it is clearly visible that SVM achieves better recognition accuracy firstly with MIFS and secondly with the CIFE among all the tested feature selection algorithms.

The performance of MIFS is evaluated with *k* = 30, 50 or 80 selected indices of cumulant features as input to the SVM classifier. Figure 10 shows the performance variation of MIFS with *k* features for different datasets for β = 0.8. With an increase in *k*, the performance of MIFS increases in all environments. MIFS achieves better performance in the home environment for all *k* (30/50/80). The classification and recognition task reports better recognition accuracy for β = 0.5 and 0.8. MIFS achieves better recognition accuracy with *k* = 80, in all environments. Compared to the home environment, MIFS reports slightly lesser performance in a lab environment and further under performs in lab + home environment. In the lab environment, MIFS performance is better in a dataset with 276 gestures obtained from 1 user than the dataset with 150 gestures obtained from multiple users. For 10 × 5 cross-validation, MIFS performs better than all feature selection algorithms with an overall higher accuracy of 97.84%, 98.18% and 96.34% for lab, home, and lab + home environment respectively for 276 gestures with 8280 instances from one user. For 150 gestures with 7500 instances from five users in the lab environment MIFS achieves an overall higher accuracy of 96.23%. The recognition accuracy slightly declines for the lab data with 150 gestures as it involves multiple users for capturing the gesture instance, and the number of gesture samples per gesture per user decreases.

HOS-Re achieves better recognition accuracy with β = 0.8 for *k* = 80 in all environments, as in Figure 10. Hence, to study the influence of β on the recognition accuracy with MIFS, experiments were carried out by systematically varying β from 0.1 to 1. Figure 11 highlights the variation in accuracy with β for different datasets. MIFS achieves better recognition accuracy with increasing β value as it reduces the uncertainty between the features and gesture class. Home dataset (276 gestures) shows higher accuracy than dataset from the lab (276 gestures). Lab + home showed the least accuracy levels compared to other datasets in 276 gestures category, for all β. Lab and home dataset (276 gesture) show consistent accuracy levels at β = 0.2 and above. Interestingly, lab dataset shows a mildly decreasing trend in accuracy with an increase in β. For Lab + home, the peak recognition accuracy of 96.34% observed at β = 0.5. MIFS captures the arbitrary mutual relations that exist between the individual third-order cumulant feature without performing any transformation. Therefore, MIFS provides the robust subset of cumulant features as the input to the multi-level SVM classifier preserving the mutual information with less uncertainty.

Figure 12 shows the comparison of overall recognition accuracy of CIFE algorithm with *k* on different datasets. Similar to MIFS, the accuracy of CIFE increases with *k* and shows higher recognition accuracy for the home dataset.

Table 3 compares the overall recognition accuracy of the proposed methodology HOS-Re with corresponding values from the literature [29] adopting CNN with and without signal processing (SP). The number of users (*N_u_*) that participated in the lab and home environment data acquisition with 276 gestures is 1, whereas for a lab environment with 150 gestures is 5. The number of samples per gesture per user (*N_su_*) for which the CSI values were acquired is 20 and 10 for lab and home environments with 276 gestures (*N_g_*). For a lab environment with 150 gestures, this value is 10. The product of *N_u_* × *N_su_* × *N_g_* gives the total number of gesture samples (*N_gs_*) acquired. From this *N_gs_*, the number of gesture samples per user is estimated to be 5520 and 2760 for lab and home environment respectively with 276 gestures (lab 276 and home 276). For the lab environment with 150 gestures (lab 150), this value is 1500. 

SignFi [29] reported 95.72% recognition accuracy for lab 276 and for home 276, it is 93.98% when no signal pre-processing is applied. This reduction in recognition accuracy could be attributed to the reduced samples per user with home 276. Deep learning methods adopted in [29] is expected to produce higher gesture recognition accuracy with an increase in sample size. However, the combined lab + home 276 datasets reported 92.21% recognition accuracy. This drop in accuracy is anticipated due to the influence of environmental factors in lab 276 dataset, as inferred from Figure 6a.

SignFi [29] with signal pre-processing achieves recognition accuracies higher than the corresponding values without pre-processing. For lab 276 the accuracy is 98.01%, and for home 276 it is 98.91%. This marginal increase in home 276 can be due to lower interference in the home environment compared to the *lab* environment. As observed earlier, the recognition accuracy of lab + home 276 declines to 94.81%. Lab 150 achieves least recognition accuracy of 86.66% since the number of samples per user acquired is lesser than other datasets, where the performance of deep learning method suffers. 

Deep learning methods [29] had large variations in recognition accuracy estimates influenced by the interferences and/or sample size. The present work HOS-Re achieved consistently higher recognition accuracies in all scenarios. In particular, when the samples acquired per user is lesser as in *lab* 150, the recognition accuracy estimated by present work is 96.23%, and it is almost 10% higher than the corresponding value obtained from the deep learning methods with signal pre-processing. It is preferred to have a gesture recognition method achieving higher recognition accuracy with minimum samples, as it is challenging to acquire a large number of samples from human participants. Therefore, the presently adopted machine learning approach along with the identified feature extraction, selection, and classification methods perform well, irrespective of the sample size and environmental factors.

For more insight on the robustness of recognition performance of the presently adopted HOS-Re methodology, the results from the present work are compared with the related work reported in the literature. Table 4 compares some of the related work on gesture recognition using COTS Wi-Fi devices, and it can be observed that most of the work reported so far used lesser samples. SignFi [29] is the only dataset having more number of gesture classes and samples instances. Amongst the cited work in Table 4, the works that used machine learning classifiers on CSI traces to recognize the gesture, have not handled such sizeable data and complex multi-class classification task. It can also be observed that the presently adopted HOS-Re methodology achieves the highest recognition accuracy.

#### 5.4.2. SVM Classifier Execution Time

The computational complexity is a key index for evaluating a given algorithm. Figure 13 shows the average time per sample for training and testing on each dataset, using MIFS with *k* = 80. The time taken for testing phase is higher than the training phase for all datasets. Lab dataset with 150 gestures reports least execution time per sample (Training: 2.76 ms; Testing: 3.13 ms), whereas *lab* + *home* dataset with 276 gestures reports highest execution time (Training: 4.05 ms; Testing: 8.94 ms). The simulations are done on an Intel(R) Core(TM) i5-7200U CPU @2.50GHz 2.71GHz RAM 8.00 GB.

## 6. Conclusions

HOS-Re adopts a ML approach and proposes higher order statistical features that extracts useful information from the non-Gaussian signal distribution. Results from presently proposed HOS-Re approach, achieve attractive classification accuracy with MIFS on SignFi dataset [29]. HOS-Re extracts third-order cumulant feature that maximizes the recognition accuracy. Subsequently, feature selection methods derived from information theory construct a robust and highly mutually informative feature subset as input to multilevel SVM classifier with Gaussian RBF kernel to measure the performance. MIFS and CIFE report better performance compared to other feature selection algorithms. However, MIFS reports the most striking performance in comparison to CIFE in extracting the mutually informative cumulant feature subset. The importance of reducing the uncertainty and preserving useful information considering the non-linear relationship for achieving better recognition accuracy is evident while looking at the other feature selection criteria in the present study.

MIFS reports the best overall trade-off accuracy because it selects highly mutually informative cumulant feature subset and it significantly reduces the uncertainty among the features and the gesture class. For 10 × 5 cross-validations, HOS-Re with MIFS achieved average recognition accuracy of 97.84%, 98.26% and 96.34% for the *lab*, home and *lab* + *home* environment respectively. The average recognition accuracy for 150 sign gestures with 7,500 instances, collected from five different users was 96.23% in the lab environment. This work presents clear evidence that with proper feature extraction and selection technique, ML algorithms can achieve notable recognition accuracy, even without applying any form signal processing technique.

## Figures and Tables

**Figure 1 sensors-19-02959-f001:**
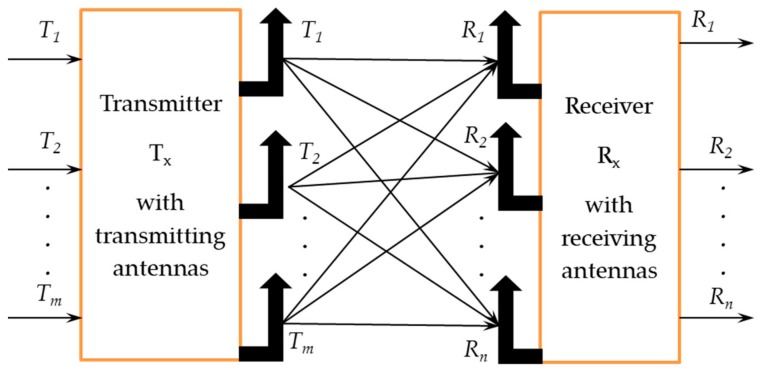
CSI representation of MIMO.

**Figure 2 sensors-19-02959-f002:**
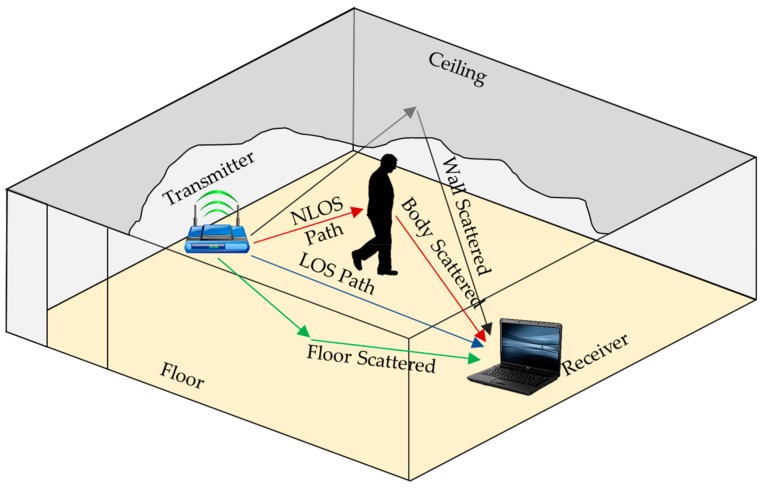
Wi-Fi signal propagation in an indoor environment.

**Figure 3 sensors-19-02959-f003:**
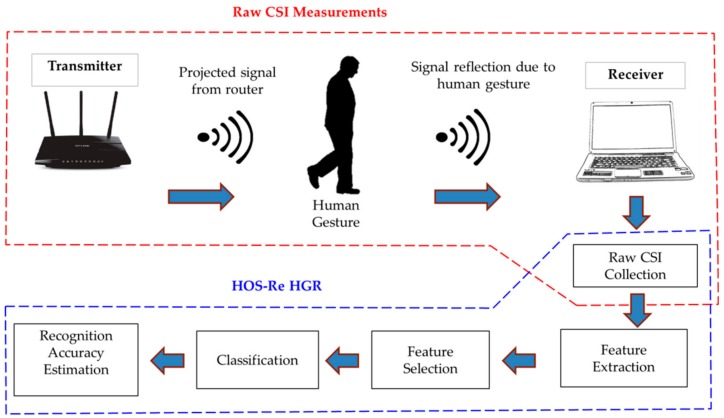
System overview.

**Figure 4 sensors-19-02959-f004:**
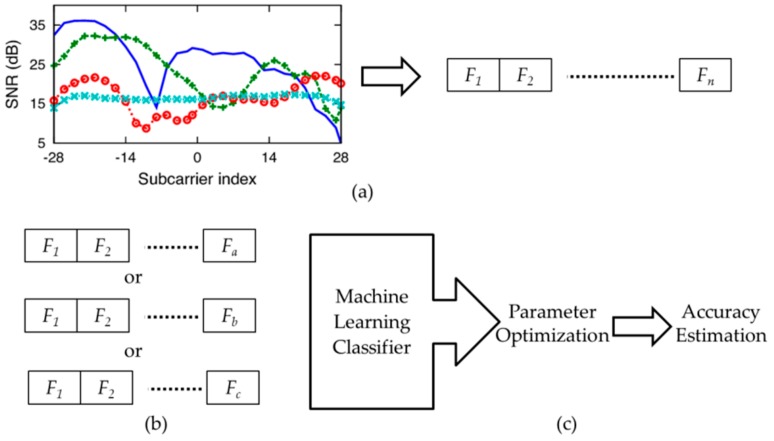
HOS-Re methodology representing (**a**) Feature extraction; (**b**) Feature selection; (**c**) Classification.

**Figure 5 sensors-19-02959-f005:**
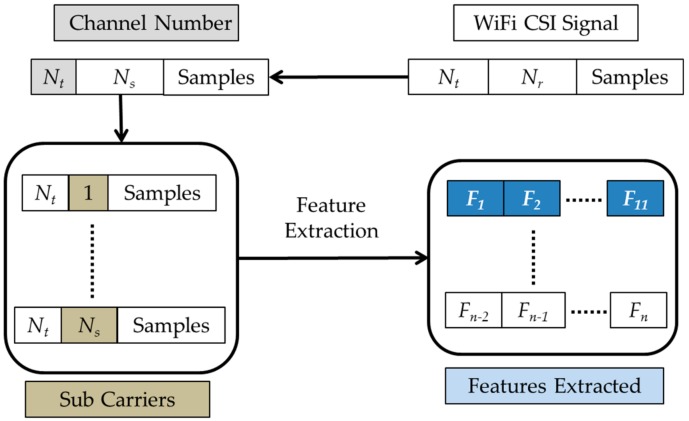
Feature extraction.

**Figure 6 sensors-19-02959-f006:**
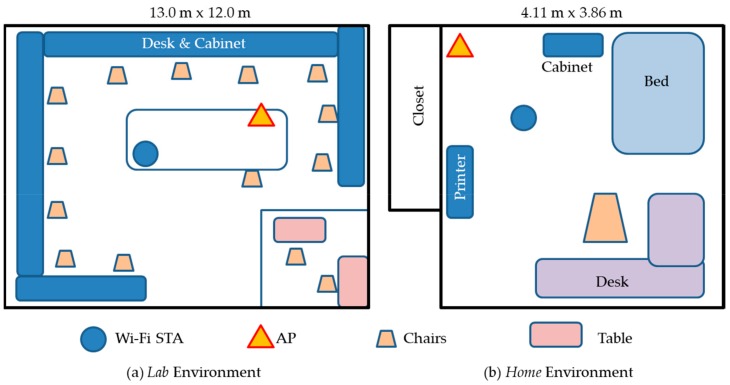
Schematic representation of testing environments as reported in SignFi [29].

**Figure 7 sensors-19-02959-f007:**
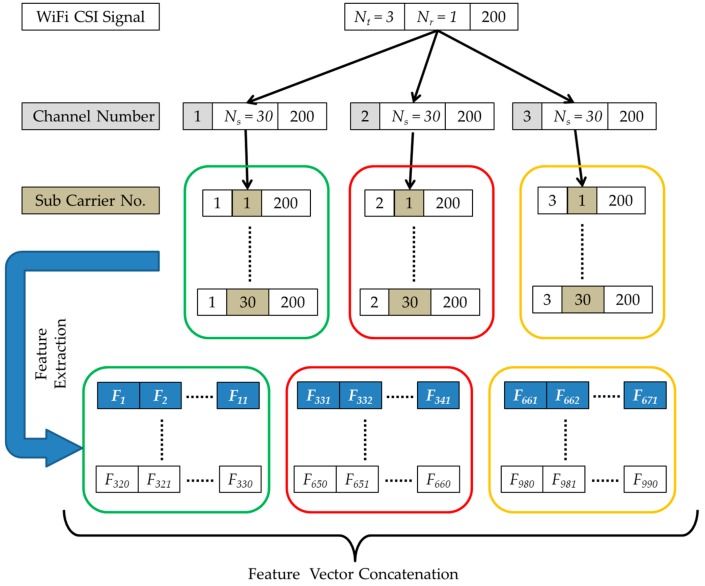
Cumulant feature extraction.

**Figure 8 sensors-19-02959-f008:**
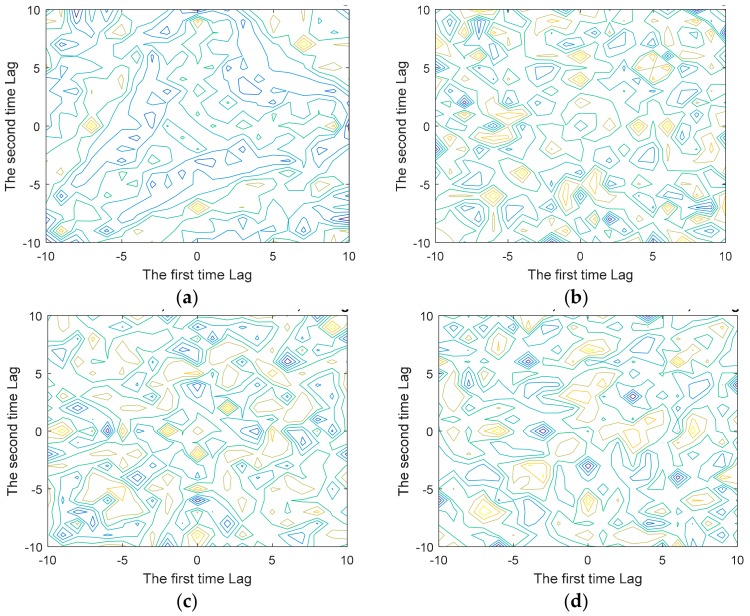
The 2D contour plot of third order cumulants (**a**) ‘HORSE’ (Sample 25); (**b**) ‘MONKEY’ (Sample 27); (**c**) ‘SNAKE’ (Sample 30); (**d**) ‘TIGER’ (Sample 31).

**Figure 9 sensors-19-02959-f009:**
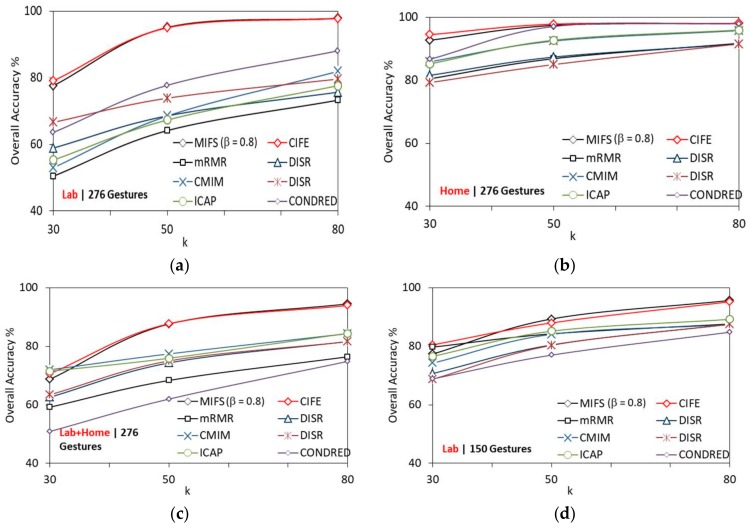
Comparison of feature selection algorithms with *k* (**a**) 276 gestures − lab dataset; (**b**) 276 gestures − home dataset; (**c**) 276 gestures − lab + home dataset; (**d**) 150 gestures − lab dataset.

**Figure 10 sensors-19-02959-f010:**
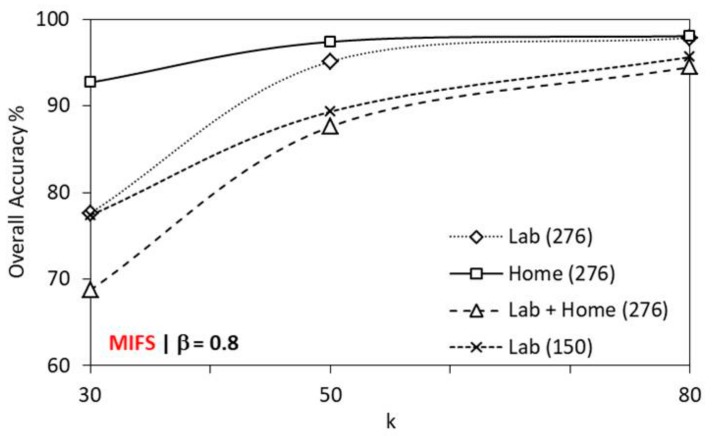
Performance of MIFS with *k* features on a different dataset.

**Figure 11 sensors-19-02959-f011:**
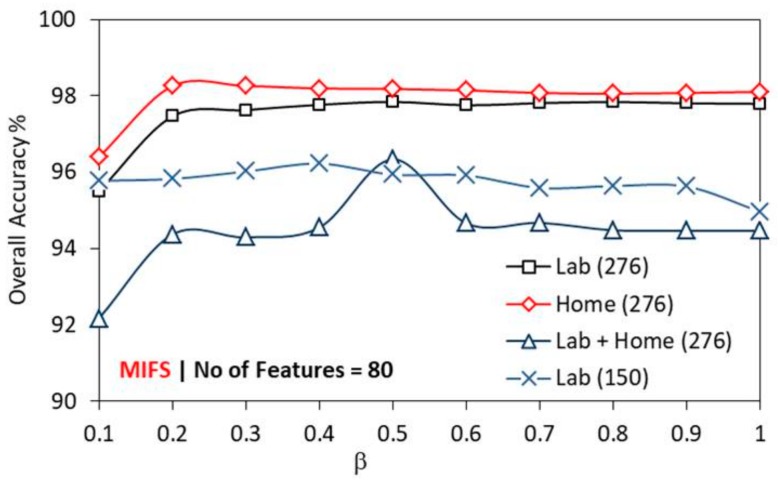
Performance of MIFS with β on different datasets.

**Figure 12 sensors-19-02959-f012:**
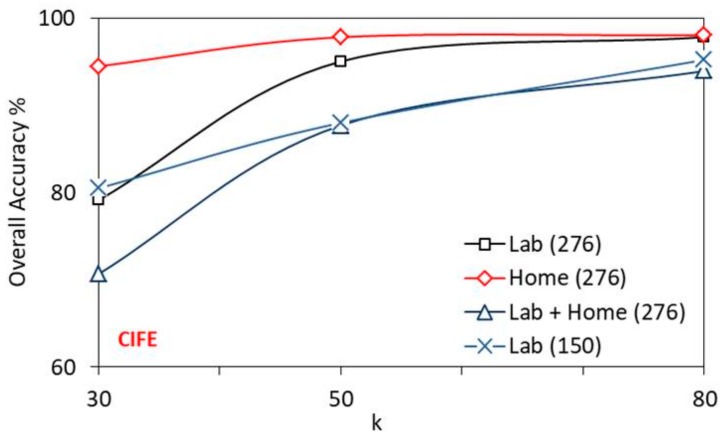
Performance of CIFE with *k* on different datasets.

**Figure 13 sensors-19-02959-f013:**
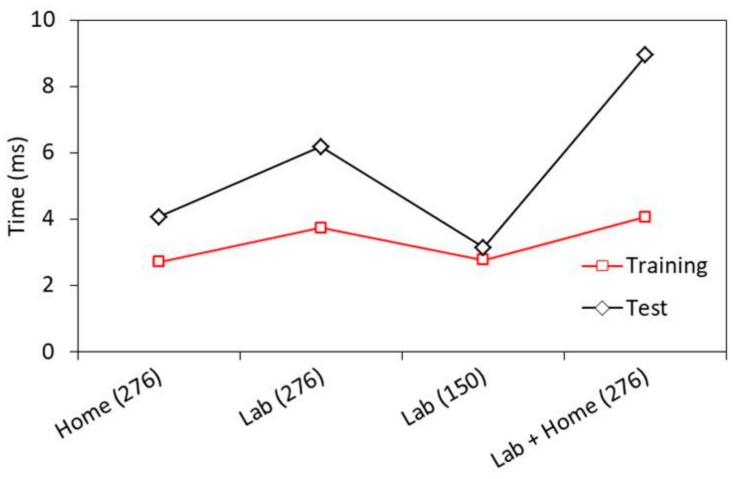
SVM classifier execution time.

**Table 1 sensors-19-02959-t001:** Gesture class representation—part of the complete dataset [29].

Gesture Class	Count of Samples	Description								
Common	16	‘FINISH’	‘GO’	‘HELP’	‘LIKE’	‘LOVE’	‘MORE’	‘NEED’	‘NO’	‘NOT’
		‘PLEASE’	‘RIGHT’	‘SORRY’	‘WANT’	‘WITH’	‘WITHOUT’	‘YES’	-	-
Animals	15	‘ANIMAL’	‘BEAR’	‘BIRD’	‘CAT’	‘COW’	‘DOG’	‘ELEPHANT’	‘GIRAFFE’	‘HORSE’
		‘LION’	‘MONKEY’	‘PET’	‘RAT’	‘SNAKE’	‘TIGER’	-	-	-
Colors	12	‘BLACK’	‘BLUE’	‘BROWN’	‘COLOR’	‘GRAY’	‘GREEN’	‘ORANGE’	‘PINK’	‘PURPLE’
		‘RED’	‘WHITE’	‘YELLOW’	-	-	-	-	-	-
Descriptions	32	‘AWFUL’	‘BAD’	‘BALD’	‘BEAUTIFUL’	‘CLEAN’	‘COLD’	‘CUTE’	‘DEPRESSED’	‘DIFFERENT’
		‘EXCITED’	‘FINE’	‘GOOD’	…	-	-	-	-	-
Family	31	‘AUNT’	‘BABY’	‘BOYFRIEND’	‘BROTHER’	‘CHILDREN’	‘COUSIN(FEMALE)’	‘COUSIN (MALE)’	‘DAUGHTER’	‘DIVORCE’
		‘DAUGHTER’	‘DIVORCE’	‘FAMILY’	‘FATHER’	‘FRIEND’	…	-	-	-
Food	54	‘DELICIOUS’	‘DINNER’	‘DRINK’	‘EAT’	‘EGGS’	‘FISH’	‘FOOD’	‘FRUIT’	‘GRAPES’
		‘HAMBURGER’	‘HOTDOG’	‘ICECREAM’	‘KETCHUP’	‘LEMON’	…	-	-	-
Home	17	APARTMENT’	‘BATHROOM’	‘BICYCLE’	‘BUY’	‘CAR’	‘CLEAN’	‘DOOR’	‘GARAGE’	‘GO-IN/ENTER’
		‘HOME’	‘HOUSE’	‘KITCHEN’	‘ROOM’	‘SHOWER’	‘SOFA’	‘TELEPHONE’	‘TOILET’	-
People	13	‘ASK-YOU’	‘BOY’	‘DEAF’	‘GIRL’	‘GIVE-YOU’	‘HEARING’	‘I/ME’	‘MAN’	‘MY/MINE’
		‘WOMAN’	‘WORK’	‘YOU’	‘YOUR’	-	-	-	-	-
Questions	6	‘HOW’	‘WHAT’	‘WHEN’	‘WHERE’	‘WHO’	‘WHY’	-	-	-
School	26	‘BOOK’	‘BUS’	‘CLASS’	‘COLLEGE’	‘DORM’	‘DRAW’	‘ELEMENTARY’	‘ENGLISH’	‘HIGH-SCHOOL’
		‘HISTORY’	‘KNOW’	…	-	-	-	-	-	-
Time	31	‘AFTER’	‘AFTERNOON’	‘AGAIN’	‘ALWAYS’	‘BEFORE’	‘DAY’	‘FRIDAY’	‘FUTURE’	‘HOUR’
		‘LAST/PAST’	‘MIDNIGHT’	…	-	-	-	-	-	-
Others	23	‘AIRCONDITIONER’	‘BODY’	‘LEG’	‘FAN’	‘RADIO’	‘TV’	REFRIGERATOR	‘BROWSER’	‘WINDOW’
		‘WASHER’	‘COMPUTER’	…	-	-	-	-	-	-

**Table 2 sensors-19-02959-t002:** Mutual information-based feature selection Algorithms.

Algorithms	Description
MIFS [20]	A greedy approach that selects only highly informative feature and forms an optimal feature subset. It identifies the non-linear relationship between the selected feature and its output class to reduce the amount of redundancy and uncertainty in the feature vector [61,62,63].
mRMR [21,22]	An incremental feature selection algorithm that forms an optimal feature subset by selecting features with minimum Redundancy and Maximum Relevancy.
CIFE [23]	It forms an optimal feature subset that maximizes the class-relevant information between the features by reducing the redundancies among the features.
JMI [24]	An increment to mutual information which finds the conditional mutual information to define the joint mutual information among the features and eliminates the redundant features if any.
CMI [25]	Selects the features only if it carries additional information and eases the prediction task of the output class.
DISR [26]	DISR measures the symmetrical relevance and combines all features variable to describe more information about the output class instead of focusing on individual feature information.
ICAP [27]	Features are selected based on the interactions and understand the regularities of the feature set.
CONDRED [28]	Identifies the conditional redundancy exists between the features.

**Table 3 sensors-19-02959-t003:** Comparison of overall recognition accuracy.

Dataset	Number of Gestures (*N_g_*)	Number of Users (*N_u_*)	Number of Samples per Gesture per User (*N_su_*)	Total Number of Gesture Samples (*N_gs_*)	Number of Samples Per User	Overall Recognition Accuracy		
						SignFi + CNN without SP [29]	SignFi + CNN with SP [29]	Present Work without SP (HOS-Re)
*Lab* 276	276	1	20	5520	5520	**95.72**	98.01	**97.84**
*Home* 276	276	1	10	2760	2760	**93.98**	98.91	**98.26**
*Lab* + *Home* 276	276	1	-	8280	8280	**92.21**	94.81	**96.34**
*Lab* 150	150	5	10	7500	1500	-	86.66	**96.23**

**Table 4 sensors-19-02959-t004:** Comparison of recognition accuracy of systems using COTS Wi-Fi devices.

Reference/ Sensing Metric	Signal Processing (SP)	Learning Algorithm	Application	Number of Gestures	Recognition Accuracy
Wigest [34]/ RSSI	Wavelet Filter; FFT, DWT; Threshold based signal extraction;	Pattern Matching	Hand Gesture Recognition: Hand movements with mobile device.	7 hand gestures	87.5%/96% (1 AP/3 AP’s)
WiGer [36]/ CSI	Butterworth low pass filter	Segmentation: multi-level wavelet decomposition algorithm and the short-time energy algorithm, DTW	Hand gesture recognition	7 hand gestures	97.28%, 91.8%, 95.5%, 94.4% and 91% (Scenario 1 to 5)
WiCatch [37]/ CSI	MUSIC algorithm	SVM	Two hand moving trajectories recognition	9 hand gesture	95%
WiFinger [33]/ RSSI and CSI	Butterworth filter, Wavelet based denoising and PCA	DTW	Finger gesture recognition	8 finger gestures	76%(RSSI) and 95% (CSI)
WiKey [15]/ CSI	Low pass filter, PCA, DWT Shape features	DTW	Keystroke recognition	37 keys	77.4% to 93.4%
Mudra [38]/ CSI	Thresholding	Stretch limited DTW	Mudra recognition	9 finger gestures	96%
SignFi [29]/ CSI	Without SP	CNN	Sign language gesture recognition	276 gestures	95.72%, 93.98%, and 92.21% for *lab* 276, *home* 276 and *lab* + *home* 276 environments respectively
SignFi [29]/ CSI	With SP—Multiple Linear Regression	CNN	Sign language gesture recognition	276 gestures	98.01%, 98.91%, 94.81%, and 86.66% for *lab* 276, *home* 276, *lab* + *home* 276 and *lab* 150 environments respectively
HOS-Re (Present work)	Without SP Cumulant Features	SVM	Sign language gesture recognition	276 gestures	97.84%, 98.26%, 96.34%, and 96.23% for *lab* 276, *home* 276, *lab* + *home* 276 and *lab* 150 environments respectively

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
