# Peer review of "Higher Order Feature Extraction and Selection for Robust Human Gesture Recognition using CSI of COTS Wi-Fi Devices"

_sensors, 2019, doi:10.3390/s19132959_

Reviewer 1 Report

Confusion matrix should be used as an additional tool to show the results.

The problem of environment-dependence of the results should be discussed at least in the Introduction Section, also referring to past works such as:

S. Di Domenico, M. De Sanctis, E. Cianca, F. Giuliano, G. Bianchi, “Exploring Training Options for RF Sensing Using CSI”, IEEE Communications Magazine, vol. 56, no. 5, May 2018.

Author Response

Point 1: Confusion matrix should be used as an additional tool to show the results.

Response 1: Thank you for your valuable suggestion. Effectively the confusion matrix is a very powerful tool to compare the results; however, we didn’t opt for it in this work because we are treating a highly complex problem. The present work adopts a multiclass problem and the numbers of the classes / gestures handled here is either 276 or 150 in number. Hence it is quite challenging to present the confusion matrix in a table of 276 x 276 or 150 x 150 in the article.

Point 2: The problem of environment-dependence of the results should be discussed at least in the Introduction Section, also referring to past works such as:

S. Di Domenico, M. De Sanctis, E. Cianca, F. Giuliano, G. Bianchi, “Exploring Training Options for RF Sensing Using CSI”, IEEE Communications Magazine, vol. 56, no. 5, May 2018.

Response 2:  Thank you for your valuable suggestions. We agree with your assessment to discuss about the problem of environmental dependence of the results in the introduction section. The impact of results due to environmental factors is mentioned in P2, 49 and the recommended article provides valuable perspective, therefore we have cited the article as [14] in the revised manuscript as follows:

It is evident that the following factor like number of users [10] and Access Point (AP) [11], orientation and distance between the users, and transmitter and receiver pair [12,13], environmental factors [14], interferences, and multipath fading effect in the sensing environment influence the recognition accuracy [15].

Reviewer 2 Report

The paper uses raw channel state information to recognize human gesture, and a higher order statistics based recognition model containing amplitude, phase information and signal to noise ratio information is produced. The recognition performance is evaluated by using a ML classifier. However, the experiment data come from a public database SignFi, and the accuracy of recognition results is lower than that of cited SignFi [28]. There is no analysis for the difference. It is suggested that more experiments be added to further verify the reliability of the method.

P4, 177: the Channel State Information (CSI) in 3.1 section is basic concepts, it should be described briefly.

P8, 274: the third order cumulants consider SNR, What is the role of SNR? How is it obtained? 

P8, 286: the paper extract 11 cumulant feature, Please elaborate on what the 11 features specifically mean.

P8, 298: the size k as 30, 50 and 80, How to choose this size, please explain it.

P9, 309: Generally, 10 cross-validation is used in machine learning. Why a 10×5-cross-validation is adopted for performance evaluation, can you provide some insight?

P9, 319: Implementation on SignFi Dataset in 5.1 section, What is the reason to choose this dataset? The figures of measurement setting of the lab and home environments are missing.

P12, 336: the recognition accuracy of human gesture using the cumulant feature extraction should be analyzed, and recognition performance comparisons with other existing feature extraction methods should also be analyzed. 

P13, 366: the methods of feature selection in 5.3 section are existing methods, only need a brief statement.

P16, 442: about Performance of MIFS with k features on different dataset in figure 8, Why does the value of the x coordinate only choosing 30, 50 and 80, Why not consider other values, such as k less than 30 or greater than 80?

P18, 489: In this paper, HOS-Re achieved an average recognition accuracy of 97.84%, 98.26% and 96.34% for the lab, home and lab+home environment respectively, but the average recognition accuracy of cited SignFi[28] is 98.01%, 98.91%, and 94.81% for the lab, home, and lab+home environment, respectively. Can you explain why some of this article's recognition accuracy is lower?

Author Response

Dear Reviewer,

Thank you for the valuable comments. Attached is the response for your perusal.

Reviewer 3 Report

This paper presents a method for hand gesture recognition using commodity WIFi cards. Authors propose higher order statistics based recognition where HOS features are extracted from CSI traces and processed. Results demonstrate the ability to classify gestures from existing datasets that are publicly available 

My main concerns with the work are:

- Hand gesture recognition is an extensively studied problem, with as authors noted many systems achieving a very high > 97% accuracy. The work presented by authors achieve similar accuracy, so I am unsure of the scientific novelty of the work. The results don't look at all exciting to me.

- Authors propose  FS methods but never motivate the reason for using these methods over other states of the artworks.

- The paper is verbose and very very difficult to follow.

I have the following suggestions to the authors to improve the quality of the manuscript:

+ The manuscript uses too many shorthand notations (CF, CSI, RBF etc), which makes it very difficult to follow the text.

+ In the introduction and conclusion the author mentions a remarkable result. It will be nice if authors could quantify this result

+ it is very difficult to understand the gestures, as they are not visualised. It will be nice if authors could visualise some of the gestures detected. 

Author Response

Dear reviewer,

Thank you for your valuable comments. Attached is the response for your perusal.

Round  2

Reviewer 1 Report

Fine as it is

Reviewer 2 Report

With the revised version of the manuscript, and considering the authors' comment, all of the problems which I have brought up have been solved. I think the manuscript with this format is eligible to be published.